# Scaling Supervision for Free: Leveraging Universal Segmentation Model for Enhanced Medical Image Diagnosis

**Yingtai Li**[1,2]                                                    LIYINGTAI@MAIL.USTC.EDU.CN
**Shuai Ming**[3]
**Haoran Lai**[1,2]
**Fenghe Tang**[1,2]
**Wei Wei**[3]
**S Kevin Zhou**[1,2,4,5]                                                S.KEVIN.ZHOU@GMAIL.COM

[1] *School of Biomedical Engineering, Division of Life Sciences and Medicine, University of Science and Technology of China (USTC), Hefei Anhui, 230026, China*

[2] *Center for Medical Imaging, Robotics, Analytic Computing & Learning (MIRACLE), Suzhou Institute for Advance Research, USTC, Suzhou Jiangsu, 215123, China*

[3] *The First Affiliated Hospital of USTC, Division of Life Sciences and Medicine, USTC, Hefei, Anhui, 230001, China*

[4] *State Key Laboratory of Precision and Intelligent Chemistry, USTC, Hefei, Anhui 230026, China*

[5] *Key Laboratory of Intelligent Information Processing of Chinese Academy of Sciences (CAS), Institute of Computing Technology, CAS, Beijing, 100190, China*

**Editors:** Accepted for publication at MIDL 2025

## Abstract

Deep learning-based medical image analysis has been constrained by the limited availability of large-scale annotated data. While recent advances in large language models have enabled scaling automatic extraction of diagnostic labels from reports, we propose that scaling other form of supervision could be an equally important yet unexplored direction. Inspired by the success of foundation models, we leverage modern universal segmentation model to scale anatomical segmentation as an additional supervision signal during training. Through extensive experiments on three large-scale CT datasets totaling 58K+ volumes, we demonstrate that incorporating this "free" anatomical supervision consistently improves the performance of various mainstream architectures (ResNet, ViT, and Swin Transformer) by up to 12.74%, with particularly significant gains for Transformer-based models and anatomically-localized abnormalities, while maintaining inference efficiency as the segmentation branch is only used during training. This work opens up a new scaling direction for medical imaging and demonstrates how existing universal segmentation models can be repurposed to enhance diagnostic models at virtually no additional cost.

**Keywords:** supervision scaling, CT diagnosis

## 1. Introduction

Precise computer-aided diagnosis (CAD) is one of the main goals of medical image analysis. In the deep learning era, building performant medical image diagnosis systems usually involves curating large-scale datasets paired with diagnostic labels. Scaling paired data with automatically extracted labels has greatly enlarged the scale of available datasets in the past few years (Hamamci et al., 2024; Draelos et al., 2021; Irvin et al., 2019; Wang et al.,

2017; Cid et al., 2024), which pave the way for the development of many successful models (Li et al., 2018; Liu et al., 2019; Hamamci et al., 2024; Kim et al., 2021; Pham et al., 2021; Yan et al., 2018; Van Sonsbeek et al., 2023). While scaling data is a common technique to improve model performance, it faces unique challenges in medical contexts - certain diseases have fixed occurrence rates, which fundamentally limits our ability to obtain more training samples (James et al., 2018; Mitani and Haneuse, 2020). This constraint has led researchers to explore alternative approaches for improving model performance.

Scaling supervision, rather than just scaling data, has emerged as a promising direction. Recent work has demonstrated several successful approaches to this end. For example, concept bottleneck models (Koh et al., 2020; Tan et al., 2025; Gao et al., 2024) explicitly learn interpretable intermediate "concepts" that can enhance both performance and model interpretability. The success of these approaches shows that incorporating additional forms of supervision can be as valuable as increasing dataset size in improving model performance, which also leads to more explainable models.

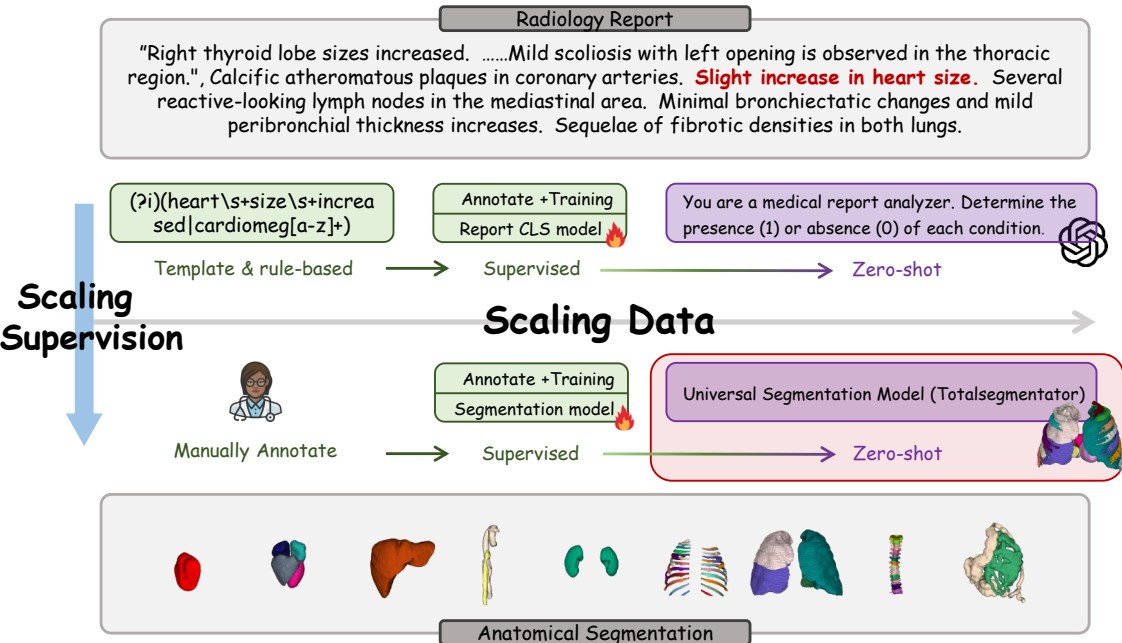

Figure 1: Our approach parallels using natural language processing algorithms to automate label extraction from medical reports, offering a cost-effective way to scale data, and takes an orthogonal direction to scale the type of supervision.

In this work, we propose localization information such as anatomical segmentation as another valuable direction for supervision scaling. Localization information is crucial for diagnosis, as it provides spatial context for lesions and enhances explainability. The success of introducing localization information to CAD systems has been validated by a wide range of research works and prize winning competition practices (Li et al., 2018; Liu et al., 2019; Lin et al., 2023; Ardila et al., 2019; Cao et al., 2023; Wang et al., 2024; Jiménez-Sánchez

et al., 2018). This can be provided through various forms such as bounding boxes (Li et al., 2018; Irvin et al., 2019; Nguyen et al., 2022), detected landmarks (Lin et al., 2023), segmentation masks (Rudie et al., 2024), and signed distance maps (Zheng et al., 2024), in either one-stage or two-stage pipelines (Lin et al., 2023; Wang et al., 2024; Cao et al., 2023; Ardila et al., 2019). While traditional approaches require manual annotation of localization information, which is prohibitively expensive given the expertise required, we explore whether modern segmentation models can automatically provide this valuable supervision signal.

Our approach draws inspiration from recent developments in automated label generation. In particular, the evolution of extracting diagnostic labels from medical reports provides an interesting parallel. Initially, this process relied on manually designed templates and rule-based natural language processing (NLP) methods (Draelos et al., 2021; Wang et al., 2017; Irvin et al., 2019). It then progressed to supervised approaches using deep learning models such as BERT (Devlin, 2018) to extract labels (Hamamci et al., 2024; Bustos et al., 2020), and most recently to more efficient methods using large language models (LLMs) (Park et al., 2024). This progression shows how automated approaches can effectively scale diagnostic label while reducing manual effort. Despite the limitation of NLP algorithms and the imperfect quality of medical reports, many successful models have been trained with dataset constructed from mining free-text reports and shown promising results in more serious validation and clinical practice (Cid et al., 2024).

We foresee a similar opportunity with anatomical segmentation. Medical image segmentation has been extensively studied for decades, with modern models achieving remarkable performance across various anatomies, especially for the CT modality. While other forms of localization information such as bounding boxes, landmarks, or distance maps can be valuable, their development lags behind, with no universal model readily available, universal segmentation models trained on large-scale datasets (Wasserthal et al., 2023; Li et al., 2024; Ma et al., 2024; Zhao et al., 2024; Ren et al., 2024) are readily available. We hereby ask: Similar to how NLP models now automate report label extraction, *can universal segmentation model provide valuable supervision signals to boost the performance of medical image analysis models?*

To answer this question, we conduct extensive experiments. Using three large-scale CT datasets (Hamamci et al., 2024; Draelos et al., 2021; Rudie et al., 2024) totaling around 60,000 volumes, we generate segmentation masks using TotalSegmentator (Wasserthal et al., 2023; Isensee et al., 2021) and evaluate how adding automatically generated segmentation supervision impact the performance of various mainstream vision backbones (ResNet (He et al., 2016), ViT (Dosovitskiy, 2020), and Swin Transformer (Liu et al., 2021)). We use video variants (Liu et al., 2022; Fan et al., 2021; Tran et al., 2018) of these models provided by torchvision offical implementation. To isolate the effect of adding segmentation supervision, we keep our approach as simple as possible - using only a linear layer for pixel-level classification at the lowest resolution feature maps, followed by upsampling to the original image size for loss computation.

Our experiments demonstrate that these *automatically generated segmentation masks provide significant performance improvements without requiring additional manual annotations*, especially for Transformer-based models and abnormalities with fixed anatomical locations. We hope that our findings can facilitate the development of more accu-

rate and explainable medical image analysis systems. Code will be available at https://github.com/SigmaLDC/AutoSeg-Scale-Supervision.

In summary, we introduce the following contributions:

- We propose anatomical segmentation as a new direction for scaling supervision in medical image analysis, demonstrating that automatically generated anatomical masks can provide effective auxiliary supervision without any additional manual annotation cost

- We conduct comprehensive empirical analysis using three large-scale CT datasets (58K+ volumes) across multiple anatomical regions, providing strong empirical evidence that scaling supervision consistently improves diagnostic across multiple mainstream architectures, with particularly significant gains for Transformer-based models

- We introduce a simple yet effective approach for incorporating segmentation supervision that requires no architectural changes and adds zero inference overhead, making it readily applicable to existing medical image analysis pipelines

## 2. Methods

### 2.1. Automatic segmentation mask generation

We use TotalSegmentator (Wasserthal et al., 2023; Isensee et al., 2021) to generate segmentation masks for all CT volumes in our experiments. TotalSegmentator is a deep-learning based segmentation model that can segment over 100 anatomical structures from CT volumes. The model is a suite of nnUNet (Isensee et al., 2021) models trained on a large dataset of manually annotated CT scans and has shown robust performance across different scanners and protocols.

For preprocessing, we first resample all CT volumes to a common spacing of 1.5mm × 1.5mm × 3.0mm using a trilinear interpolation and stick to this spacing for training diagnosis models. The intensity values are clipped to [-1000, 1000] Hounsfield Units. We use the fast version of TotalSegmentator to accelerate the inference and save memory comsuption. The segmentation masks are saved as one-channel masks in NIfTI format with the same dimensions as the input volumes.

### 2.2. Classification training with segmentation supervision

We incorporate automatically generated segmentation masks as an auxiliary task for training the classification model. Formally, given an input CT volume $x \in \mathbb{R}^{H \times W \times D}$, our model $f_\theta$ produces feature maps $F = f_\theta(x) \in \mathbb{R}^{c \times h \times w \times d}$, where $h, w, d$ are the spatial dimensions of the lowest resolution feature map and $c$ is the number of channels.

**Classification branch:** We apply global average pooling to get the feature $z = \mathrm{GAP}(F) \in \mathbb{R}^c$ for classification, followed by a linear classifier and a normalizing function to yield the probability:

$$p_{\mathrm{cls}} = \sigma(W_{\mathrm{cls}}z + b_{\mathrm{cls}}). \tag{1}$$

For CT-RATE (Hamamci et al., 2024) and RAD-ChestCT (Draelos et al., 2021) datasets, which involve only binary multi-label classification, $\sigma$ is a sigmoid function. For RATIC

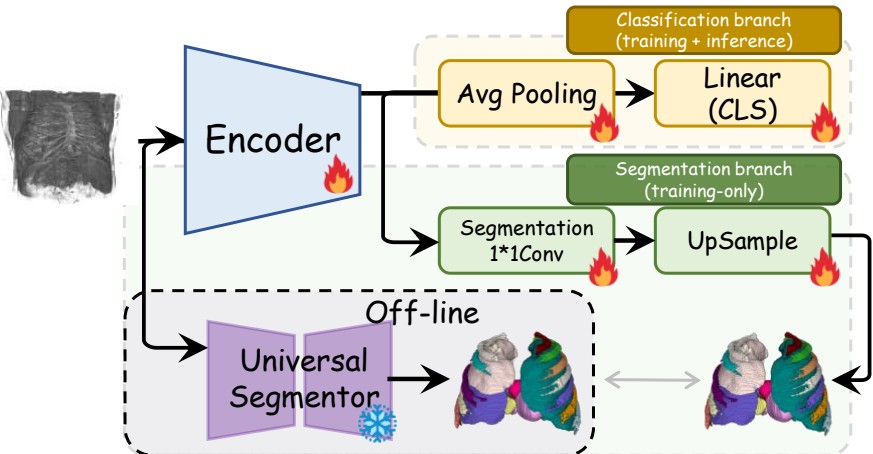

Figure 2: We generate segmentation masks using TotalSegmentator offline, and use them as an auxiliary supervision for training the classification model. For inference, only the classification branch is needed.

dataset (Rudie et al., 2024), which involves both multi-class and binary classification tasks, we use softmax for organ-specific injury grades and sigmoid for binary labels. Additionally, For the "any_injury" label in RATIC dataset, we calculate its probability as:

$$p_{\text{any\_injury}} = 1 - \prod_{\substack{i \in \{\text{bowel,liver,} \\ \text{spleen,kidney,extravasation}\}}} p_{\text{healthy}}^{(i)}. \tag{2}$$

**Segmentation branch:** To isolate the effect of adding segmentation supervision, we keep our way of introducing segmentation supervision as simple as possible to show its effectiveness, which simply add a segmentation head consisting of a linear layer ($1 \times 1 \times 1$ convolution) followed by trilinear upsampling to the original image size for loss computation:

$$p_{\text{seg}} = \text{Upsample}(W_{\text{seg}}F + b_{\text{seg}}). \tag{3}$$

The total loss varies by dataset. For binary multi-label tasks (CT-RATE and RAD-ChestCT):

$$\mathcal{L} = \mathcal{L}_{\text{cls\_1}} + \mathcal{L}_{\text{seg}}. \tag{4}$$

For RATIC, which includes multiple classification objectives::

$$\mathcal{L} = \mathcal{L}_{\text{cls\_1}} + \mathcal{L}_{\text{cls\_2}} + \lambda \mathcal{L}_{\text{any\_injury}} + \mathcal{L}_{\text{seg}}, \tag{5}$$

where $\mathcal{L}_{\text{cls\_1}}$ and $\mathcal{L}_{\text{any\_injury}}$ is binary cross entropy (BCE) loss for binary multi-label classification and $\mathcal{L}_{\text{cls\_2}}$ is the cross entropy (CE) loss for multi-class classification, $\mathcal{L}_{\text{seg}}$ is the CE loss for segmentation, and $\lambda$ is a weighting factor. While we use CE loss for segmentation in our main experiments, our preliminary experiments suggest that using Dice loss or a combination of Dice and CE loss could achieve similar performance improvements.

As TotalSegmentator predicts 128 different anatomies, predicting all these structures is computationally expensive and unnecessary. For CT-RATE and RAD-ChestCT, we use segmentation mask of the 5 lung lobes and the heart. For RATIC, we use segmentation mask of the evaluated organs: liver, spleen, left kidney, right kidney, and bowel (representing a combination of esophagus, stomach, and duodenum, small bowel, and colon).

## 3. Experiments

### 3.1. Datasets and implementation details

**Datasets:** We evaluate our method on three large-scale CT datasets: CT-RATE (Hamamci et al., 2024), RAD-ChestCT (Draelos et al., 2021), and RATIC (Rudie et al., 2024).

**CT-RATE** (Hamamci et al., 2024) consists of 25,692 chest CT scans from 21,304 unique patients, expanded to 50,188 through various reconstructions. The dataset is enriched with 18 distinct abnormalities extracted from radiology reports. CT-RATE features diverse pathologies related to heart and lung conditions, with CT scans acquired from multiple manufacturers using various imaging parameters, ensuring representation of real-world clinical heterogeneity.

**RAD-ChestCT** (Draelos et al., 2021) comprises 36,316 non-contrast chest CT volumes from Duke University Health System (2012-2017). The dataset contains 83 abnormality labels in total. Currently, only 10% (3,630 volumes) is publicly available, and we utilize these public volumes with the official split of 2,646 training and 984 test volumes, focusing on 16 classes that overlap with CT-RATE.

**RATIC** (Rudie et al., 2024) (RSNA Abdominal Traumatic Injury CT) is the largest publicly available adult abdominal trauma CT dataset, containing 4,274 studies from 23 institutions across 14 countries. We use the publicly available training set of 3,147 studies (4,711 image series), randomly split into 2,500 training and 647 test cases.

**Implementation details:** All CT volumes are resampled to a common spacing of 1.5mm × 1.5mm × 3.0mm using trilinear interpolation. The intensity values are clipped to [-1000, 200] for chest CT and [-150, 250] for abdomen CT, then normalized to [-1, 1]. Volumes are first padded/cropped to 240×240×120 or 256×256×192 voxels, then randomly cropped during training to 192×192×96 or 176×192×160 voxels. For evaluation we center crop the CT volume to 192×192×96 or 176×192×160 voxels.

For CT-RATE and RAD-ChestCT, we use equal weights for positive and negative samples. For RATIC, the sample weights are as follows: 1 for all healthy labels; 2 for low grade solid organ injuries (liver, spleen, kidney) and bowel injuries; 4 for high grade solid organ injuries; and 6 for extravasation and the auto-generated any_injury label. As the successive multiplication of the $p_{\text{any\_injury}}$ results in large gradients, we set the weight of $\mathcal{L}_{\text{any\_injury}}$ to 0.1 to stabilize gradients. The weights for segmentation loss are set to 1 for all tasks and models.

We use the AdamW optimizer with a cosine learning rate scheduler. More implementation details can be found in Appendix A.

### 3.2. Classification from scratch

We evaluate the effectiveness of segmentation supervision for different vision backbones. For each backbone, we compare performance with and without segmentation supervision.

Table 1: Classification performance (AUC-ROC %) comparison with and without segmentation supervision. Results are shown for models with and without pretrained weights initialization. Numbers in parentheses show absolute improvement from adding segmentation supervision.

| Model | CT-RATE | | RAD-ChestCT | | RATIC | |
|---|---|---|---|---|---|---|
| | w/o seg | w/ seg | w/o seg | w/ seg | w/o seg | w/ seg |
| **Without Pretrained Weights** | | | | | | |
| ResNet | 83.84 | 85.45 (+1.61) | 72.93 | 74.77 (+1.84) | 72.36 | 79.75 (+7.39) |
| MViT | 80.87 | 82.94 (+2.07) | 69.54 | 72.40 (+2.86) | 65.18 | 77.92 (+12.74) |
| Swin3D | 80.22 | 83.07 (+2.85) | 68.85 | 72.99 (+4.14) | 65.39 | 74.68 (+9.29) |
| **With Pretrained Weights** | | | | | | |
| ResNet | 84.83 | 85.64 (+0.81) | 74.81 | 75.93 (+1.12) | 82.56 | 84.51 (+1.95) |
| MViT | 82.13 | 84.36 (+2.23) | 72.01 | 73.90 (+1.89) | 70.91 | 81.92 (+11.01) |
| Swin3D | 82.30 | 85.58 (+3.28) | 72.32 | 75.01 (+2.69) | 70.88 | 80.55 (+9.67) |

The results in Table 1 demonstrate that adding segmentation supervision consistently improves classification performance across all backbones and datasets. The improvement ranges from 0.81% to 12.74% in terms of AUC-ROC, with larger gains observed on Transformer-based models and relatively smaller datasets, which may hint more supervision is especially useful for training Transformer-based models, possibly due to their data-hungry nature and the challenge of learning effective attention patterns from limited samples.

Furthermore, we observe that, for the abnormalities strongly correlated with specific anatomical regions, our method exhibits larger improvements, suggesting that the anatomical knowledge from segmentation is particularly helpful for these categories. A comprehensive results of all categories with more metrics are provided in Appendix B.

### 3.3. Classification with pretrained weights

Leveraging pretrained weights as initialization is widely adopted practice in medical image analysis, as they can provide a good starting point for the training of a model, especially when the scale of data and annotation is limited. To validate the practical value of adding segmentation as an auxiliary supervision signal, we also evaluate the performance of the classification model with pretrained weights as initialization. The improvements remain substantial even with pretrained weights, indicating that segmentation supervision provides unique benefits beyond what is captured in pretraining.

### 3.4. Analysis of training dynamics

We track the gradient norm of the models across training iterations and the corresponding training loss, the results are shown in Figure 3. We observe the gradient gradually increase during the training process, while the curvature remains relatively stable, and the training loss experience an accelerated decrease. This phenomenon, combined with dete-

Analysis of Training Dynamics

Figure 3: Training dynamics comparison between resnet3D with and without segmentation supervision. The bold line represents time weighted average. A more flat minima is observed with segmentation supervision.

riorating test performance, suggests the model is converging towards sharp minima in the loss landscape. In contrast, models trained with segmentation supervision exhibit more stable gradient and smoother loss curves throughout training. The additional supervision appears to guide optimization towards flatter minima, which are generally associated with better generalization (Goodfellow, 2016). We hypothesize that the anatomical knowledge provided by segmentation supervision helps constrain the solution space, steering the model towards parameters that capture meaningful anatomical relationships rather than spurious correlations.

### 3.5. Ablation of segmentation weighting factor

We investigate the effect of varying loss weighting factors for segmentation supervision, with results presented in Table 2. The macro AUC remains relatively stable across different weighting factors. However, we observe that specific weighting factors influence performance differently across individual categories, the detailed results are provided in Appendix C.

Table 2: Impact of loss weighting factor on classification performance (AUC-ROC %).

| Segmentation Weighting Factor | 0.0 | 0.1 | 1.0 | 5.0 | 10.0 |
|---|---|---|---|---|---|
| Swin3D (RATIC) | 70.88 | 72.70 | 80.55 | 79.15 | 80.81 |

### 3.6. Impact of imperfect segmentation masks

To evaluate robustness to imperfect segmentation, we simulate common segmentation errors by applying controlled erosion/dilation operations to the segmentation boundaries. These operations mimic typical failure modes in automated segmentation. Our analysis reveals that the proposed method maintains stable performance under light to moderate boundary perturbations. The detailed results are provided in Appendix C.

### 3.7. Ablation on anatomy sensitivity

Our main experiments utilize segmentation of liver, kidney, spleen, and bowel—anatomies directly related to the target abnormalities. To assess anatomical specificity, we conduct additional experiments using segmentation of individual organs. Results show that liver and spleen segmentation alone yield minimal improvements, while kidney and bowel segmentation provide substantial gains.

Table 3: Impact of different anatomical segmentations on performance (AUC-ROC %).

| Model | RATIC | | | | | |
|---|---|---|---|---|---|---|
| | No segmentation | Liver only | Kidney only | Spleen only | Bowel only | All organs |
| Swin3D | 70.88 | 69.15 | 76.15 | 71.17 | 79.60 | 80.55 |

## 4. Limitations

Despite consistent improvements across datasets and models, our approach has several limitations. First, it depends on the availability of robust universal segmentation models for the target modality. While such models exist for CT and increasingly for MRI, they are less developed for other imaging modalities like ultrasound or nuclear medicine, limiting immediate cross-modality applicability. Second, our findings indicate that not all anatomical segmentations contribute equally to performance improvements. The boundary conditions that determine when and how anatomical supervision proves beneficial require further investigation to fully characterize the method's scope and optimal application scenarios.

## 5. Conclusion and discussion

In this work, we demonstrate that segmentation masks automatically generated by modern universal segmentation models can serve as effective auxiliary supervision signals for training CT diagnosis models. Our comprehensive experiments across three large-scale datasets show consistent performance improvements across different architectures and tasks, particularly for Transformer architectures and abnormalities with strong anatomical correlations. The challenge of obtaining high-quality supervision has long been a bottleneck in medical image analysis, where expert annotation is both expensive and time-consuming. Our approach offers a practical solution by leveraging existing universal segmentation models to provide "free" anatomical supervision. This parallels the successful use of natural language processing to automate diagnostic label extraction from medical reports, suggesting a broader paradigm of repurposing mature AI models to scale supervision in medical imaging.

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

## Appendix A. More implementation details

### A.1. Class used in RAD-ChestCT dataset

We list the class used in RAD-ChestCT dataset in Table 4.

Table 4: Class used in RAD-ChestCT Dataset

| Class | | |
|---|---|---|
| 1. cathether_or_port | 2. lymphadenopathy | 3. spetal_thickening |
| 4. hernia | 5. nodule | 6. pericardial_effusion |
| 7. scarring | 8. bronchial_wall_thickening | 9. opacities |
| 10. emphysema | 11. cardiomegaly | 12. pleural_effusion |
| 13. bronchiectasis | 14. consolidation | |
| 15. atelectasis | 16. calcification | |

### A.2. Data preprocessing details

We list the data preprocessing details in Table 5. Different intensity clipping ranges and crop sizes are applied to chest and abdomen CT scans. A larger crop size is used for abdomen CT, as we observed that the abdomen CT scans in the RATIC dataset typically had larger dimensions compared to the chest CT scans in the CT-RATE and RAD-ChestCT datasets. This size difference likely stems from the fact that many abdomen CT scans in the dataset actually encompass full-body scans.

Table 5: Data Preprocessing Details

| Step | Details |
|---|---|
| Resampling | 1.5mm × 1.5mm × 3.0mm (trilinear interpolation) |
| Intensity Clipping | [-1000, 200] HU chest CT; [-150, 250] HU abdomen CT |
| Normalization | min-max normalization to [-1, 1] |
| Pad_or_Crop | 240×240×120 chest CT; 256×256×192 abdomen CT |
| Training Crop Size | 192×192×96 chest CT; 176×192×160 abdomen CT |
| Evaluation Crop | Center crop to same dimensions as training |

### A.3. Training configuration

The training configuration parameters are summarized in Table 6. The base learning rates are selected from the range of 1e-4 to 1e-5. We observe that Transformer-based models typically require lower learning rates, possibly due to their larger parameter count. The batch sizes are determined based on GPU memory constraints.

### A.4. Model architecture details

The model architecture specifications are detailed in Table 7. We use the implementation and pretrained weights of these models from the official torchvision repository, the

Table 6: Training Configuration

| Parameter | Value |
|---|---|
| Optimizer | AdamW |
| Base Learning Rate | 1e-4 (ResNet/MViT chest CT) |
| | 5e-5 (Swin chest CT) |
| | 1e-5 (MViT/Swin RATIC) |
| Weight Decay | 1e-4 |
| LR Schedule | Cosine decay |
| Batch Size | *Chest CT:* |
| | 10 (ResNet) |
| | 6 (Swin) |
| | 4 (MViT) |
| | *Abdomen CT:* |
| | 5 (ResNet) |
| | 2 (Swin, MViT) |
| Training Epochs | 20 (CT-RATE) |
| | 100 (RAD-ChestCT/RATIC) |
| Loss Functions | BCE with Logits Loss (binary) |
| | Cross Entropy Loss (multi-class) |
| Gradient Clipping | Maximum norm of 5.0 |

pretrained weights are trained on the Kinetics-400 dataset. We modify the patch embedding layer (or stem layer) of these models to 1-channel, and do not use the corresponding pretrained weights for these layers. This is because the pretrained weights are trained on RGB videos, which has 3 channels, and our CT scans are single-channel. For models with positional encoding, we modify the positional encoding to match the input size.

Table 7: Model Architecture Details

| Model | Modifications |
|---|---|
| ResNet | • Modified R3D-18 input layer to single-channel 3D convolution (kernel=$2\times2\times2$, stride=2) |
| Swin | • Modified Swin3D-T with custom patch embedding (patch_size=$16\times2\times2$, in_channels=1, embed_dim=96) |
| MViT | • Custom 3D projection layer (kernel=$16\times2\times2$, stride=$16\times2\times2$) |
| | • Modified MViT-v2-S with adjusted positional encoding (spatial_size=$96\times96$, temporal_size=6) |

## A.5. Evaluation metrics

We adopt the Area Under the Receiver Operating Characteristic curve (AUC-ROC) as our primary evaluation metric, supplemented by additional metrics to ensure comprehensive performance assessment. The optimal threshold is selected based on the ROC-based optimal threshold, and we use this threshold to calculate all metrics.

Table 8: Performance Metrics (%) Comparison: Models With and Without Segmentation Supervision. Numbers in parentheses show absolute improvement from adding segmentation supervision.

| Model | Metric | CT-RATE | | RAD-ChestCT | | RATIC | |
|---|---|---|---|---|---|---|---|
| | | w/o seg | w/ seg | w/o seg | w/ seg | w/o seg | w/ seg |
| **Without Pretrained Weights** | | | | | | | |
| ResNet | AUC | 83.84 | 85.45 (+1.61) | 72.93 | 74.77 (+1.84) | 72.36 | 79.75 (+7.39) |
| | Sens | 78.49 | 80.03 (+1.54) | 69.05 | 70.44 (+1.39) | 67.87 | 74.52 (+6.65) |
| | Spec | 78.17 | 78.83 (+0.66) | 78.77 | 73.09 (-5.68) | 70.35 | 75.04 (+4.69) |
| | F1 | 79.77 | 80.64 (+0.87) | 75.71 | 75.14 (-0.57) | 82.40 | 84.46 (+2.06) |
| | Prec | 43.58 | 45.47 (+1.89) | 43.56 | 40.40 (-3.16) | 45.35 | 46.85 (+1.50) |
| MViT | AUC | 80.87 | 82.94 (+2.07) | 69.54 | 72.40 (+2.86) | 65.18 | 77.92 (+12.74) |
| | Sens | 75.98 | 78.55 (+2.57) | 67.40 | 68.04 (+0.64) | 63.43 | 74.79 (+11.36) |
| | Spec | 74.98 | 75.57 (+0.59) | 67.60 | 71.00 (+3.40) | 66.42 | 73.85 (+7.43) |
| | F1 | 77.23 | 78.47 (+1.24) | 71.64 | 73.85 (+2.21) | 85.00 | 80.05 (-4.95) |
| | Prec | 40.03 | 41.84 (+1.81) | 36.04 | 39.05 (+3.01) | 42.38 | 45.64 (+3.26) |
| Swin3D | AUC | 80.22 | 83.07 (+2.85) | 68.85 | 72.99 (+4.14) | 65.39 | 74.68 (+9.29) |
| | Sens | 75.20 | 77.37 (+2.17) | 66.18 | 69.98 (+3.80) | 63.11 | 72.29 (+9.18) |
| | Spec | 74.66 | 76.77 (+2.11) | 67.16 | 70.47 (+3.31) | 64.33 | 72.33 (+8.00) |
| | F1 | 77.09 | 79.00 (+1.91) | 70.37 | 73.29 (+2.92) | 81.27 | 78.87 (-2.40) |
| | Prec | 39.14 | 42.53 (+3.39) | 34.75 | 38.96 (+4.21) | 41.16 | 44.39 (+3.23) |
| **With Pretrained Weights** | | | | | | | |
| ResNet | AUC | 84.83 | 85.64 (+0.81) | 74.81 | 75.93 (+1.12) | 82.56 | 84.51 (+1.95) |
| | Sens | 78.88 | 80.12 (+1.24) | 70.28 | 69.96 (-0.32) | 74.67 | 78.31 (+3.64) |
| | Spec | 77.52 | 79.73 (+2.21) | 75.48 | 72.43 (-3.05) | 80.59 | 80.11 (-0.48) |
| | F1 | 80.07 | 81.05 (+0.98) | 76.76 | 74.45 (-2.31) | 81.64 | 86.04 (+4.40) |
| | Prec | 44.37 | 46.08 (+1.71) | 41.81 | 41.55 (-0.26) | 46.51 | 50.10 (+3.59) |
| MViT | AUC | 82.13 | 84.36 (+2.23) | 72.01 | 73.90 (+1.89) | 70.91 | 81.92 (+11.01) |
| | Sens | 76.82 | 78.38 (+1.56) | 67.71 | 69.66 (+1.95) | 67.55 | 77.49 (+9.94) |
| | Spec | 76.59 | 77.81 (+1.22) | 71.76 | 70.63 (-1.13) | 69.14 | 77.28 (+8.14) |
| | F1 | 78.54 | 79.84 (+1.30) | 73.60 | 73.33 (-0.27) | 79.25 | 83.75 (+4.50) |
| | Prec | 41.91 | 43.86 (+1.95) | 38.11 | 39.38 (+1.27) | 43.03 | 47.04 (+3.91) |
| Swin3D | AUC | 82.30 | 85.58 (+3.28) | 72.32 | 75.01 (+2.69) | 70.88 | 80.55 (+9.67) |
| | Sens | 76.62 | 79.92 (+3.30) | 67.60 | 70.15 (+2.55) | 66.78 | 75.56 (+8.78) |
| | Spec | 76.80 | 79.48 (+2.68) | 70.00 | 71.40 (+1.40) | 68.84 | 76.34 (+7.50) |
| | F1 | 78.71 | 81.14 (+2.43) | 72.65 | 74.67 (+2.02) | 80.62 | 82.63 (+2.01) |
| | Prec | 41.58 | 45.96 (+4.38) | 37.51 | 39.91 (+2.40) | 42.57 | 47.16 (+4.59) |

**Note:** AUC = Area Under ROC Curve, Sens = Sensitivity, Spec = Specificity, Prec = Precision. All metrics are macro-averaged across classes.

# Appendix B. Detailed results

We provide results with more metrics in Table 8. We also provide the results with all 18 abnormality categories in CT-RATE dataset in Table 10, the 16 abnormality categories in RAD-ChestCT dataset in Table 11, and the 14 abnormality categories in RATIC dataset in Table 12.

Table 9: Sensitivity and specificity comparison with and without segmentation pretraining on CT-RATE dataset. Detailed results with all 18 abnormality categories.

| Model | Metric | CT-RATE | | | | | | | | | | | | | | | | | | |
|---|---|---|---|---|---|---|---|---|---|---|---|---|---|---|---|---|---|---|---|---|
| | | AWC | ATL | BRE | CDM | CON | CAC | EMP | HH | IST | LN | LOP | LAP | MM | MAP | PBT | PCE | PLE | PFS | AVG |
| R | Sens | 87.43 | 70.13 | 78.18 | 89.23 | 87.26 | 87.45 | 66.67 | 73.86 | 84.74 | 61.65 | 78.04 | 65.65 | 76.68 | 83.79 | 78.87 | 80.08 | 94.41 | 68.71 | 78.49 |
| | Spec | 84.07 | 71.11 | 76.67 | 85.56 | 80.39 | 80.83 | 73.56 | 73.27 | 79.79 | 69.73 | 78.87 | 73.56 | 80.08 | 79.18 | 70.49 | 83.97 | 93.09 | 72.92 | 78.17 |
| R + S | Sens | 90.08 | 70.55 | 72.43 | 90.15 | 87.44 | 86.93 | 75.50 | 80.10 | 87.95 | 67.89 | 76.01 | 66.42 | 79.87 | 84.19 | 74.37 | 80.53 | 94.68 | 64.14 | 80.03 |
| | Spec | 84.35 | 75.54 | 71.76 | 85.30 | 83.08 | 82.76 | 75.98 | 76.20 | 78.17 | 64.24 | 80.27 | 73.78 | 87.13 | 77.60 | 76.42 | 88.38 | 94.48 | 63.45 | 78.83 |
| R + P | Sens | 86.62 | 72.23 | 70.91 | 88.00 | 87.09 | 89.94 | 77.50 | 73.14 | 87.15 | 67.74 | 75.08 | 71.10 | 79.55 | 76.29 | 77.47 | 80.53 | 93.09 | 66.43 | 78.88 |
| | Spec | 85.50 | 73.99 | 70.80 | 87.25 | 79.82 | 86.54 | 68.51 | 77.19 | 79.39 | 63.05 | 78.28 | 68.58 | 81.44 | 80.73 | 70.19 | 79.95 | 93.73 | 70.33 | 77.52 |
| R + P + S | Sens | 89.85 | 73.77 | 69.70 | 91.70 | 86.06 | 89.42 | 73.50 | 75.06 | 87.55 | 70.24 | 78.46 | 69.33 | 82.75 | 81.82 | 80.85 | 81.86 | 94.15 | 66.07 | 80.12 |
| | Spec | 85.36 | 73.73 | 75.86 | 87.21 | 79.66 | 86.59 | 76.51 | 79.06 | 78.21 | 66.15 | 79.89 | 74.67 | 84.74 | 84.14 | 76.60 | 87.53 | 95.61 | 63.63 | 79.73 |

Table 10: Classification performance (AUC-ROC %) comparison with and without segmentation pretraining on CT-RATE dataset. Detailed results with all 18 abnormality categories.

| Model | CT-RATE | | | | | | | | | | | | | | | | | | |
|---|---|---|---|---|---|---|---|---|---|---|---|---|---|---|---|---|---|---|---|
| | AWC | ATL | BRE | CDM | CON | CAC | EMP | HH | IST | LN | LOP | LAP | MM | MAP | PBT | PCE | PLE | PFS | AVG |
| R | 90.1 | 77.3 | 79.2 | 94.3 | 88.3 | 88.3 | 80.2 | 79.3 | 87.8 | 68.1 | 86.3 | 76.8 | 87.1 | 89.8 | 80.9 | 89.2 | 97.3 | 68.9 | 83.84 |
| R + S | 92.3 | 79.3 | 79.9 | 94.7 | 89.9 | 91.9 | 81.2 | 83.4 | 88.0 | 69.9 | 86.8 | 78.8 | 89.4 | 89.6 | 82.1 | 91.9 | 97.2 | 71.8 | 85.45 |
| R + P | 93.0 | 79.3 | 78.5 | 92.5 | 89.9 | 93.4 | 79.9 | 83.0 | 87.9 | 69.8 | 85.4 | 77.4 | 89.5 | 87.8 | 81.6 | 89.0 | 97.3 | 71.8 | 84.83 |
| R + P + S | 93.1 | 79.6 | 80.9 | 94.5 | 90.3 | 93.6 | 79.9 | 85.9 | 87.8 | 69.2 | 86.2 | 77.7 | 91.3 | 88.4 | 82.5 | 91.3 | 97.1 | 72.5 | 85.64 |

**Note:** Model variants: R (ResNet), R + S (ResNet+seg), R + P (ResNet+pretrain), R + P + S (ResNet+pretrain+seg).

**Abbreviations:** AWC (Arterial Wall Calc.), ATL (Atelectasis), BRE (Bronchiectasis), CDM (Cardiomegaly), CON (Consolidation), CAC (Coronary Art. Calc.), EMP (Emphysema), HH (Hiatal Hernia), IST (Interlobular Sept. Thick.), LN (Lung Nodule), LOP (Lung Opacity), LAP (Lymphadenopathy), MM (Medical Material), MAP (Mosaic Atten. Pattern), PBT (Peribronchial Thick.), PCE (Pericardial Eff.), PLE (Pleural Eff.), PFS (Pulm. Fibrotic Seq.).

Table 11: Classification performance (AUC-ROC %) comparison with and without segmentation pretraining on RAD-ChestCT dataset. Detailed results with all 16 abnormality categories.

| Model | RAD-ChestCT | | | | | | | | | | | | | | | | |
|---|---|---|---|---|---|---|---|---|---|---|---|---|---|---|---|---|---|
| | COP | LAP | ST | HER | NOD | PCE | SCR | BWT | OPC | EMP | CDM | PLE | BRE | CON | ATL | CAL | AVG |
| R | 74.12 | 75.07 | 82.57 | 65.31 | 64.40 | 69.13 | 64.05 | 66.25 | 64.31 | 80.94 | 85.34 | 92.97 | 74.16 | 74.27 | 67.37 | 67.73 | 72.93 |
| R + S | 84.25 | 71.14 | 71.80 | 63.76 | 73.55 | 64.81 | 59.85 | 62.92 | 85.34 | 87.55 | 86.32 | 93.00 | 77.56 | 75.76 | 71.84 | 72.24 | 74.77 |
| R + P | 94.55 | 67.88 | 81.12 | 64.52 | 67.21 | 63.16 | 63.51 | 65.10 | 62.07 | 81.19 | 87.91 | 92.76 | 75.63 | 73.24 | 69.98 | 76.89 | 74.81 |
| R + P + S | 91.46 | 75.70 | 86.07 | 72.41 | 69.60 | 72.28 | 65.71 | 60.63 | 62.33 | 85.89 | 87.81 | 93.30 | 73.48 | 74.32 | 67.68 | 76.16 | 75.93 |

**Note:** Model variants: R (ResNet), R + S (ResNet+seg), R + P (ResNet+pretrain), R + P + S (ResNet+pretrain+seg).

**Abbreviations:** COP (Cathether or Port), LAP (Lymphadenopathy), ST (Septal Thickening), HER (Hernia), NOD (Nodule), PCE (Pericardial Effusion), SCR (Scarring), BWT (Bronchial Wall Thickening), OPC (Opacities), EMP (Emphysema), CDM (Cardiomegaly), PLE (Pleural Effusion), BRE (Bronchiectasis), CON (Consolidation), ATL (Atelectasis), CAL (Calcification).

Table 12: Classification performance (AUC-ROC %) comparison with and without segmentation pretraining on RATIC dataset. Detailed results with all 14 abnormality categories.

| Model | RATIC | | | | | | | | | | | | | | |
|---|---|---|---|---|---|---|---|---|---|---|---|---|---|---|---|
| | AI | BH | BI | EH | EI | KH | KHI | KL | LH | LHI | LL | SH | SHI | SL | AVG |
| R | 72.0 | 69.3 | 69.3 | 65.0 | 65.0 | 74.1 | 76.2 | 69.2 | 73.4 | 84.5 | 69.4 | 75.7 | 80.9 | 69.0 | 72.36 |
| R + S | 77.4 | 84.1 | 84.1 | 68.6 | 68.6 | 84.6 | 82.1 | 86.5 | 77.0 | 92.6 | 71.0 | 80.7 | 88.7 | 70.4 | 79.75 |
| R + P | 80.8 | 82.3 | 82.3 | 73.4 | 73.4 | 88.8 | 87.5 | 87.3 | 78.3 | 89.6 | 74.2 | 86.1 | 91.3 | 80.3 | 82.56 |
| R + S + P | 78.8 | 93.2 | 93.2 | 76.4 | 76.4 | 91.8 | 89.7 | 81.7 | 79.0 | 94.4 | 73.6 | 85.4 | 90.3 | 79.8 | 84.51 |

**Note:** Model variants: R (ResNet), R + S (ResNet+seg), R + P (ResNet+pretrain), R + P + S (ResNet+pretrain+seg).

**Abbreviations:** AI (Any injury), BH (Bowel healthy), BI (Bowel injury), EH (Extravasation healthy), EI (Extravasation injury), KH (Kidney healthy), KHI (Kidney high), KL (Kidney low), LH (Liver healthy), LHI (Liver high), LL (Liver low), SH (Spleen healthy), SHI (Spleen high), SL (Spleen low).

## Appendix C. More ablation results

### C.1. Ablation on segmentation weighting factor

In Table 13, we show the classification performance with different segmentation weighting factors on RATIC dataset. We observe that the performance is relatively stable with a wide range of segmentation weighting factors, while different segmentation weighting factors may favor different abnormality categories.

Table 13: Classification performance (AUC-ROC %) with different segmentation weighting factors on RATIC dataset. Detailed results with all 14 abnormality categories.

| Weights | RATIC | | | | | | | | | | | | | | |
|---|---|---|---|---|---|---|---|---|---|---|---|---|---|---|---|
| | AI | BH | BI | EH | EI | KH | KHI | KL | LH | LHI | LL | SH | SHI | SL | AVG |
| 0 | 70.40 | 73.25 | 73.25 | 62.23 | 62.23 | 69.57 | 72.49 | 67.45 | 70.81 | 75.92 | 68.69 | 68.36 | 70.88 | 65.53 | 70.88 |
| 0.1 | 71.69 | 83.94 | 83.94 | 70.41 | 70.41 | 70.78 | 69.94 | 69.33 | 69.08 | 79.70 | 65.09 | 72.35 | 72.35 | 68.91 | 72.70 |
| 1.0 | 77.34 | 84.61 | 84.61 | 71.52 | 71.52 | 82.80 | 86.62 | 81.48 | 78.92 | 96.57 | 73.07 | 78.86 | 85.00 | 71.26 | 80.55 |
| 5.0 | 77.11 | 83.64 | 83.64 | 75.47 | 75.47 | 81.96 | 83.07 | 81.64 | 73.05 | 88.62 | 68.28 | 79.65 | 85.98 | 70.55 | 79.15 |
| 10.0 | 78.26 | 84.44 | 84.44 | 74.25 | 74.25 | 86.88 | 85.33 | 88.04 | 76.81 | 91.39 | 71.49 | 78.91 | 86.35 | 70.53 | 80.81 |

**Abbreviations:** AI (Any injury), BH (Bowel healthy), BI (Bowel injury), EH (Extravasation healthy), EI (Extravasation injury), KH (Kidney healthy), KHI (Kidney high), KL (Kidney low), LH (Liver healthy), LHI (Liver high), LL (Liver low), SH (Spleen healthy), SHI (Spleen high), SL (Spleen low).

### C.2. Ablation on impact of imperfect segmentation masks

We conduct experiments on the RATIC dataset to evaluate the impact of imperfect segmentation masks on classification performance, the result is shown in Table 14. A distortion factor of x means x % of boundary pixels will gone through a erosion/dilation operation. From the result we conclude that the performance is relatively robust against imperfect segmentation masks. While since segmentation mask is not used during inference, we suspect a significant failure of segmentation of inference image would not lead to a significant drop of performance.

Table 14: Impact of imperfect segmentation masks on classification performance (AUC-ROC %).

| Distortion Factor | 0 | 10 | 50 |
|---|---|---|---|
| Swin3D (RATIC) | 80.81 | 80.36 | 81.30 |

