# OpenReview forum: "Scaling Supervision for Free: Leveraging Universal Segmentation Models for Enhanced Medical Image Diagnosis"
_MIDL.io/2025/Conference — MIDL 2025 Poster_

### Official Review · Reviewer_3jUC · 2025-02-16

**Confidence:** 4
**Preliminary Rating:** 4
**Recommendation:** Poster
**Final Rating:** 4

**Summary:**

This paper proposes leveraging a popular pre-trained segmentation model to generate anatomical labels in CT volumes which may be used in a multi-task training setting to boost the performance of a standard classification model. Authors evaluate this method on a number of backbone architectures and different CT datasets, showing consistent performance improvements and empirical results that suggest favourable training dynamics in the multi-task setting.

**Strengths:**

• Extensive ablation across architectures: Comparing ResNet, ViT and Swin architectures gives confidence that the proposed method’s improvements are consistent across different choices of architecture. Use of multiple large-scale datasets provides strong evidence of the proposed method’s performance.
• Insightful Training Dynamics: The analysis of gradient norms and loss curves offers a deeper understanding of how anatomical supervision guides optimisation toward flatter minima, which is associated with improved generalization.

**Weaknesses:**

• Incremental Novelty: While the idea of using auxiliary tasks is well-known in multi-task learning, the novelty here lies in the “free” aspect of using a pre-trained segmentation model. However, more discussion on how this supervision differs qualitatively from existing multi-task approaches would be beneficial.
• Lack of code: There is no code released with the paper to reproduce their experiments,  however authors mention in the paper that they intend to release code.

**Detailed Comments:**

• Segmentation Quality Impact: It would be useful for the authors to discuss potential failure modes when TotalSegmentator produces suboptimal segmentation masks and how that might affect diagnostic performance.
• Ablation on Anatomical Structures: More detailed ablation studies on the choice and number of anatomical structures used for supervision could clarify their individual contributions.
• Out-of-domain robustness: I’m not sure it would be possible with any of the given datasets, but it would be very valuable to understand how the segmentation supervision affects model robustness between datasets (e.g. trained on CT-RATE -> tested on Rad-Chest).
• Qualitative interpretation - it would be nice if class activation maps or attention maps were shown for a few examples, to show whether the segmentation-guided models’ maps are better localised on the relevant anatomy.
• Figure 1 took me a while to comprehend - on first glance it looks like it is comparing the raw radiological reports to anatomical segmentation data as the raw data. It may be better to clarify that the top half of the figure is concerned with image-level classification, while the bottom half is concerned with pixel-level segmentation.
• Add a GitHub repo! :)

**Justification Of The Final Rating:**

Thank you to the authors for their addressing of the various points. The primary concern, and reason for choosing a 4 instead of a 5, was the limited novelty in what is a (well-implemented and evaluated) multi-task learning with only one task carried forward to inference. This concern still stands - it is not clear that there is any 'specific optimisation' as the authors refer to. One could equally e.g. throw away the decoder of a U-Net at inference if the U-Net was trained as a multi-task learner with a classification loss on the bottleneck.

**Justification Of The Preliminary Rating:**

This paper presents a simple and practical approach to enhancing medical image diagnosis by repurposing universal segmentation models to provide auxiliary anatomical supervision during training. The method is innovative in that it scales the type of supervision rather than merely the amount of data, and it manages to boost performance significantly (up to 12.74% improvement in AUC-ROC) across several large-scale CT datasets and diverse model architectures, particularly benefiting Transformer-based models. While the approach is well-validated empirically and supported by insightful analyses of training dynamics, the novelty of the work is limited. This justifies a weak accept recommendation.

**Questions To Address In The Rebuttal:**

• Robustness to Segmentation Errors: How does the method handle cases where the segmentation model fails or produces noisy masks?
• Sensitivity Analysis: Can the authors provide more details on the sensitivity of performance gains to the selection and number of anatomical structures used for supervision?
• Generalisability: Do the authors have any preliminary results or insights into extending this approach to other imaging modalities or less well-represented anatomical regions?
• Comparison with Multi-task Learning: How does the “free” supervision approach compare with conventional multi-task learning setups that include manually annotated segmentation?

**Special Issue:**

No

---

> ### Author Response · Authors · 2025-03-07
> **Response to Reviewer 3jUC**
>
> Thank you for your thoughtful review. We've addressed your points as follows:
>
> **Robustness to Segmentation Errors**
>
> Our robustness experiments demonstrate that performance remains stable with perturbation on segmentation masks. Since the segmentation branch is only used during training, inference performance isn't directly affected by segmentation failures.
>
> **Sensitivity Analysis**
>
> As you suggested, we conducted experiments using segmentation of individual organs. Results show that kidney and bowel segmentation alone provide substantial gains (76.15% and 79.60% respectively vs. 70.88% baseline), while liver and spleen segmentation yield minimal improvements.
>
> **Generalisability**
>
> While our current work focuses on CT, our approach conceptually can extend to any modality with available robust universal segmentation models. MRI applications are particularly promising as segmentation foundation models mature for this modality.
>
> **Comparison with Multi-task Learning**
>
> Unlike conventional multi-task learning approaches that aim to balance performance across multiple objectives, our method is specifically optimized to enhance classification performance. By decoupling segmentation and classification during inference, we maintain computational efficiency while benefiting from the rich anatomical priors during training. Conventional multi-task learning approaches may face scaling limitations, while our method's key advantage is its ability to scale efficiently to larger datasets.

---

### Official Review · Reviewer_QZ1n · 2025-02-17

**Confidence:** 5
**Preliminary Rating:** 3
**Recommendation:** Poster
**Final Rating:** 4

**Summary:**

The key contribution of this work is to show that combining pretrained model to generate segmentations helps to improve the classification performance. The idea is to use segmentation as an auxiliary task and compared against pseudo segmentations produced by totalsegmentator and combined with a classification task. Analysis is performed on three different lung Chest CT based classification tasks. Analysis is performed to assess training dynamics in addition to classification accuracy.

**Strengths:**

* Paper uses a simple yet conceptually meaningful idea to combine segmentation of organs together with classification of abnormalities.
* Segmentation auxiliary task relies on pseudo segmentations hence doesn't add additional manual effort.
* Results are presented for multiple public datasets.
* Paper is well written

**Weaknesses:**

* The accuracy gains seem minimal - is this significant? Also it would be more useful to see specificity and sensitivity as well to understand if the model is biased towards majority class or if it balances both classes.
* How much is the model impacted by imperfect/incorrect segmentations. Although TotalSegmentator was trained on fairly large number of cases, it may not produce best results always. It would be helpful to understand how robust the model is to poor segmentation pseudo ground truth.
* Training analysis is good but it's unclear how it translates to accuracy. For instance, could the training be stopped in earlier epochs, which would lead to training efficiency. Is there an impact of using segmentation as auxiliary task on data efficiency?

**Detailed Comments:**

The paper presents an interesting and straightforward approach to combine segmentation as an auxiliary task with classification. Whereas there are works that have used a similar concept, the presented work takes a more practical approach to this problem by combining pseudo segmentations as ground truth for segmentation. The paper is well-written. The results show performance improvement, albeit presenting more detailed results such as including specificity, sensitivity etc would be helpful.

The analysis also focuses only on the training stability but does not take it further in terms of understand what impact does adding the auxiliary task have on classification training efficiency, data efficiency etc.

Please see listed weaknesses.

**Justification Of The Final Rating:**

Thank you for carefully addressing the concerns and the additional experiment to mimic imperfect segmentations to assess impact of segmentation variations.

To clarify, I requested cases where the totalsegmentator missed organs entirely or mostly as it often does versus than erosion and dilation.  Also, by significance, I was asking for statistical tests to measure significant differences. However, I recognize this could be done for a future version of this otherwise very interesting paper! The paper still has sufficient merits and its contributions are very timely.

**Justification Of The Preliminary Rating:**

The paper presents some interesting ideas. The issues are that the accuracy gains seem small. The benefit of using the proposed approach needs to be highlighted more clearly. This is why I gave it a borderline.

**Questions To Address In The Rebuttal:**

Can you please address the items listed in weaknesses:
* The accuracy gains seem minimal - is this significant? Also it would be more useful to see specificity and sensitivity as well to understand if the model is biased towards majority class or if it balances both classes.
* How much is the model impacted by imperfect/incorrect segmentations. Although TotalSegmentator was trained on fairly large number of cases, it may not produce best results always. It would be helpful to understand how robust the model is to poor segmentation pseudo ground truth.
* Training analysis is good but it's unclear how it translates to accuracy. For instance, could the training be stopped in earlier epochs, which would lead to training efficiency. Is there an impact of using segmentation as auxiliary task on data efficiency?

**Special Issue:**

No

---

> ### Author Response · Authors · 2025-03-07
> **Response to Reviewer QZ1n**
>
> Thank you for your insightful comments. We've addressed your concerns as follows:
>
> **Significance**
>
> The improvements are consistent across all architectures and datasets, particularly significant for transformer-based models and middle-scale datasets.
>
> **Additional Metrics**
>
> As requested, we've expanded our results to include sensitivity and specificity by category in the appendix.
>
> **Robustness to Imperfect Segmentation**
>
> We evaluated this by simulating segmentation errors through controlled boundary perturbations. Even when 50% of boundary pixels are distorted, performance remains stable (81.30% vs. 80.81% baseline), showing that the result isn't critically dependent on perfect segmentation quality.
>
> **Training Efficiency**
>
> We do not observe a consistent improvement in training efficiency. The convergence speed may differ with different network architectures and datasets. However, adding segmentation supervision does not result in slower convergence in any cases. Since adding segmentation supervision can help achieve similar performance with less data, this can be seen as a kind of data efficiency.

---

> > ### Comment · Reviewer_QZ1n · 2025-03-13
> > **Addressed most of my concerns**
> >
> > Thank you for carefully addressing the concerns and the additional experiment to mimic imperfect segmentations to assess impact of segmentation variations.
> >
> > To clarify, I requested cases where the totalsegmentator missed organs entirely or mostly as it often does versus than erosion and dilation - however, I recognize this could be done for a future version of this otherwise very interesting paper! Also, by significance, I was asking for statistical tests to measure significant differences.
> >
> > I will increase my score to weak accept.

---

### Official Review · Reviewer_RMTH · 2025-02-24

**Confidence:** 4
**Preliminary Rating:** 4
**Recommendation:** Oral

**Summary:**

In this paper, the authors proposed to leverage the universal segmentation model to improve the performance of medical image diagnosis. Specifically, the classification model is trained with a dual-branch for segmentation task and classification simultaneously. The results show that this multi-task scheme improves the classification performance with a free gain (from frozen segmentation model), especially for the abnormalities with anatomical prior.

**Strengths:**

- The idea of this paper is interesting. Though multi-task learning has been studied before in this field, it was unclear in the previous studies how to generalize this idea to many applications. With the advance of universal model, the proposed approach seems to have better generalizability and efficiency (segmentation head was not used for inference).

- The evaluation is comprehensive as it was evaluated on a large-scale dataset of CT scans (58k+). The analysis on the subtypes of abnormalities was also informative.

**Weaknesses:**

Lack of discussion on the limitation: It was unclear to me if the auxiliary segmentation task would improve all the diagnosis tasks; are there any scenarios that training with the segmentation tasks hurts the model performance?

 Lack of discussion on weighting factors of the classification loss and the segmentation loss: This is also related to the previous question: I would assume the success of this method highly depends on the weighting factors between the losses of two tasks. It would be important to investigate the impact of this weighting factor to better understand how sensitive the proposed method is.

**Detailed Comments:**

As mentioned in the limitation, it would be important to perform experiments to understand the impact of the weighting factors between the segmentation and the classification losses.

Also, it would be great to have a discussion to compare this method and the previous multi-task learning methods and the limitation of the proposed approach.

**Justification Of The Preliminary Rating:**

Overall, the paper is well-written and the proposed method is presented clearly. The experiments and analyses are comprehensive and solid. This paper can be further improved by adding the discussion of limitation and the related work, and a ablation study regarding as mentioned above.

**Questions To Address In The Rebuttal:**

See above

---

> ### Author Response · Authors · 2025-03-07
> **Response to Reviewer RMTH**
>
> Thank you for your valuable feedback. We've addressed your concerns as follows:
>
> **Weighting Factors Analysis**
>
> We conducted a comprehensive ablation study on segmentation loss weighting factors as you suggested. Results show our method remains stable across a wide range of weights (1.0-10.0), with macro AUC varying only from 80.55% to 80.81% (compared to 70.88% baseline without segmentation). This demonstrates that the approach isn't overly sensitive to this hyperparameter.
>
> **Limitations Discussion**
>
> We've added a dedicated limitations section discussing scenarios where our approach might be less effective. Performance gains are anatomy-dependent, with some structures (kidney, bowel) providing substantial improvements while others (liver, spleen) yielding minimal benefits. The method also requires robust universal segmentation models for the target modality. Additionally, current universal models mostly focus on anatomies, whereas a universal model for lesions might be more helpful.
>
> **Comparison with Previous Methods**
>
> Instead of proposing a new multi-task learning framework, we focus on leveraging existing universal segmentation models to provide "free" anatomical supervision, making it more scalable.

---

### Author Rebuttal · Authors · 2025-03-07

**Rebuttal:**

We sincerely thank all reviewers for their constructive feedback. We've significantly enhanced our paper by:

- Conducting more ablation studies on segmentation weighting factors and individual anatomical structures
- Evaluating robustness to imperfect segmentation through simulation of segmentation errors
- Providing comprehensive performance metrics including sensitivity and specificity for each category
- Expanding our discussion of limitations and potential applications to other modalities

Our additional experiments show that the proposed method is robust against a wide range of segmentation weighting factors and imperfect segmentations. We to hope address your concerns.

**Supporting Material:**

/attachment/90350894d87c4d51e541e63d55fa935af319daea.zip

---

### Comment · Area_Chair_BXZt · 2025-03-08
**author's responses are available now**

Dear Reviewers,

the rebuttals are available now. Could you take a look at the author's response and update your score please?

Thank you!

---

### Meta-Review · Area_Chair_BXZt · 2025-03-19

**Recommendation:** Accept (Poster)
**Confidence:** 4

**Metareview:**

The paper proposes leveraging universal segmentation models to provide auxiliary supervision for medical image diagnosis. By incorporating anatomical segmentation as an additional training signal, the approach improves classification performance across multiple architectures and datasets. The method is validated on three large-scale CT datasets, demonstrating up to 12.74% improvement, particularly benefiting Transformer-based models and anatomically localized abnormalities.

This work presents a practical and scalable method for enhancing deep learning-based diagnosis without requiring additional manual annotations. The authors have conducted thorough ablation studies on segmentation weighting factors, robustness to imperfect segmentations, and performance across different anatomical structures. The expanded discussion on limitations and the comparison to conventional multi-task learning approaches further strengthen the paper.

While the novelty is incremental, the method is well-executed and empirically validated. The improvements, though moderate, are consistent across architectures and datasets. Given its contributions and rigorous evaluation, I recommend accepting this paper as a poster. The approach is valuable for the community and will likely generate productive discussions at the conference.